# The Isolation, Genetic Analysis and Biofilm Characteristics of *Listeria* spp. from the Marine Environment in China

**DOI:** 10.3390/microorganisms11092166

**Published:** 2023-08-27

**Authors:** Pan Mao, Yan Wang, Lingling Li, Shunshi Ji, Peijing Li, Lingyun Liu, Jinni Chen, Hui Sun, Xia Luo, Changyun Ye

**Affiliations:** National Key Laboratory of Intelligent Tracking and Forecasting for Infectious Diseases, National Institute for Communicable Disease Control and Prevention, Chinese Center for Disease Control and Prevention, Beijing 102206, China; maopan_7@163.com (P.M.); wangyan@icdc.cn (Y.W.); ling605782805@163.com (L.L.); jishunshi2017@163.com (S.J.); lipeijing@icdc.cn (P.L.); llyyyx2021@163.com (L.L.); 15034624562@163.com (J.C.); sunhui@icdc.cn (H.S.); luoxia@icdc.cn (X.L.)

**Keywords:** *Listeria monocytogenes*, whole-genome sequencing, marine environment, resistance, virulence, biofilm

## Abstract

*Listeria monocytogenes* is an important pathogen that can cause listeriosis. Despite the growing recognition of *Listeria* spp. as a foodborne and environmental pathogen, the understanding of its prevalence and characteristics of *Listeria* spp. in the marine environment remains unknown. In this study, we first investigated the genetic and phenotypic characteristics of *Listeria* species isolated in a coastal city in China. The findings revealed that the sequence type 87 (ST87) *L. monocytogenes*, a prevalent clinical and seafood strain in China, dominates in recreational beach sands and possesses a notable biofilm-forming capacity in seawater. The presence of ST87 *L. monocytogenes* in coastal environments indicates the potential health risks for both recreational activities and seafood consumption. Moreover, the ST121 isolates from sand had a versatile plasmid encoding multifunctional genes, including *uvrX* for UV resistance, *gbuC* for salt resistance, and *npx* for oxidative resistance and multiple transposases, which potentially aid in survival under natural environments. Black-headed gulls potentially facilitate the spread of *L. monocytogenes*, with similar ST35 strains found in gulls and beach sand. As a reservoir of microbes from marine environments and human/animal excrement, coastal sand would play an important role in the spread of *L. monocytogenes* and is an environmental risk for human listeriosis.

## 1. Introduction

The genus *Listeria* consists of 30 species, as listed in the List of Prokaryotic Names with Standing in Nomenclature (LPSN) [1]. *Listeria monocytogenes* is the main human pathogenic species known to cause listeriosis, a serious infection that primarily affects in elderly, pregnant women, children, and immunosuppressed patients. *L. monocytogenes* can cause septicemia, meningitis, abortion, and other complications in humans and animals, while *L. ivanovii* causes similar pathological manifestations in ruminants, except for meningoencephalitis [2,3]. Despite the limited availability of clinical and pathogenicity data for other species, investigating the ecology, genetics, and survival strategies of these species is crucial for future public health preparedness [4]. Furthermore, *L. monocytogenes* is an environmental saprophyte that occupies a wide range of habitats, including river, soil, vegetation, and food-processing environments [2]. *L. monocytogenes* has the ability to grow under harsh environmental conditions, such as high salt concentration, pH 4.5, heavy metals, and/or freezing temperature. Studies have demonstrated that this bacterium can survive on dusty sand particulates, even under conditions of 10 °C and 88% relative humidity for more than 151 days [5]. *Listeria fleishmanii* was initially isolated from cheese and has been observed to persist for more than 5 years, and sporadic occurrences of this species have been found in soil and feed [6,7]. *Listeria aquatica* was originally identified in water in the United States [8]. *L. monocytogenes* has been reported to be constantly present in the lagoon and soil around pig manure treatment plants [9].

The species *L. monocytogenes* exhibits regional heterogeneity, ecological niche preference, and genetic diversity [10,11]. *L. monocytogenes* includes four genetic lineages (I–IV) and 14 serotypes, which are further subdivided by clonal complexes (CCs) and multilocus sequence types (STs) [10]. The majority of clinical isolates belong to the lineage I strain, while food isolates are predominantly found in lineage II [10]. Serotype 4b strains have been associated with human infections worldwide, whereas the ST87 strains of serotype 1/2b have emerged as the predominant clinical clones in China [12,13]. 

The infection of *L. monocytogenes* is well documented and is known to frequently occur through the consumption of contaminated food items, such as ready-to-eat meat, soft cheese, and unpasteurized milk [14]. Seafood contaminated with pathogenic marine bacteria is a threat to human health, particularly in coastal cities [15,16]. In addition, water and soil serve as sources of *L. monocytogenes* infection [17]. A case was reported of an adult with immunodeficiency developing septic arthritis from *L. monocytogenes* after swimming in a public pool [18]. Migratory wild birds, such as the black-headed gull, inhabit a variety of aquatic environments, which can facilitate the geographical spread of *L. monocytogenes* [19]. The occurrence of *L. monocytogenes* in public access surface water along the central California coast and its occurrence in Moroccan marine environments underscores its potential as contamination sources and its implication for the human health [20,21,22,23].

To date, an epidemiological investigation and risk assessment of *L. monocytogenes* in the marine environment in China has not yet been reported. This study aimed at a preliminary exploration of the prevalence and genetic features of *Listeria* species in coastal environments of China.

## 2. Materials and Methods

### 2.1. Sand Sample Collection and Listeria spp. Isolation

A total of 769 sand samples were collected from 5 distinct bays (denoted as A/B/C/D/E, from north to south), an island (denoted as I), and a village (denoted as V) en route to the island in a coastal city of China (Figure 1A). All sampling spots were accessible places for locals and tourists. Each collected sample, approximately 200 g of sand, was securely stored in sterilized plastic bags (3M^TM^; Minneapolis, MN, USA). Prior to bacterial isolation, the sand was subjected to socking and rubbing in roughly 100 ml of phosphate-buffered saline (PBS) (Zhongshan; Beijing, China). Following the natural deposition of sand, the supernatant was transferred into a 50 mL tube for centrifugation at 4000 rpm for 10 min. The pellets were subsequently inoculated into half-Fraser and Fraser broth (Oxoid; Basingstoke, UK) for the enrichment of *L. monocytogenes* [24]. The enriched culture was then streaked on Brilliance *Listeria* Agar (Oxoid; Basingstoke, UK) and incubated at 37 °C for 48 h. The potential clones were purified on a brain heart infusion (BHI) agar plate (Oxoid; Basingstoke, UK). Colonies indicative of potential *Listeria* presence were selected for PCR identification, employing primers targeting the *Listeria* genus-specific gene *prs* (forward: *GCTGAAGAGATTGCGAAAGAA*; reverse: *CAAAGAAACCTTGGATTTGCGG*) [25]. The amplification reaction was performed with the following steps: initial denaturation at 94 °C for 4 min, followed by 32 cycles of denaturation at 94 °C for 30 s, annealing at 54 °C for 30 s, extension at 72 °C for 1 min, and culminating with a final extension at 72 °C for 10 min. *L. monocytogenes* EGD-e was used as a positive control.

### 2.2. DNA Extraction, Sequencing, Assembly, and Species Identification

The *Listeria* spp. isolates obtained from sand samples were cultured in the BHI agar plate at 37 °C for 18 h. After incubation, the bacterial cells from one agar plate were collected via scraping followed by genomic DNA extraction using a Wizard^®^ Genomic DNA Purification Kit (Promega; Madison, WI, USA). The extracted DNA was quantified using a Qubit 2.0 Fluorometer. The DNA library was constructed using the NEBNext^®^ UltraTM DNA Library Prep Kit (NEB; 240 County Road, Ipswich, UK) with Illumina technology. The whole-genome shotgun sequencing of these isolates was performed on the Illumina Hiseq PE150 platform. The raw data were filtered to obtain clean data, and then high-quality paired reads were assembled using SOAP denovo (v 2.04). Genome sequencing, raw data filtration, and clean data assembly were performed using the instruments from Novogene Co., Ltd. (Beijing, China). For species-level identification, fastANI (v 1.33) was used to calculate the average nucleotide identity (ANI) between the genomes. Prokka (v 1.14.6) was used for genome annotation tasks.

### 2.3. Core, Pan-Genome Analyses, and Phylogenetic Analyses Based on Single-Nucleotide Polymorphism (SNP)

WGS was used for the molecular characterization of *Listeria* spp. The core and pan-genomes were analyzed with the Roary pipeline using a 95% identity cutoff. The phylogenetic tree based on the core genes of *Listeria* spp. was constructed using FastTree (v 2.1.10). The sequence types were determined with in silico multilocus sequence typing (MLST). The publicly available strains from the National Center for Biotechnology Information (NCBI) database, which shared the same sequence type as the sequenced isolates in this study, were selected for SNP analysis (Appendix A). The geographic location was summarized based on each ST of strains (Appendix A). The SNP alignment was performed using Snippy (v 4.6.0), and the phylogenetic tree was produced with Gubbins (v 2.4.1). The resulting phylogeny was plotted and visualized in ChiPlot (https://www.chiplot.online/) tools.

### 2.4. Analyses of Virulence Genes, Resistance Genes, and Plasmid 

The identification of virulence- and resistance-associated genes was accomplished through local sequence alignment with the extant *Listeria* virulence database in VirulenceFinder (https://bitbucket.org/genomicepidemiology/virulencefinder_db/src/master/) and previously reported resistance genes on linux server [26,27]. In silico antibiograms and plasmids were discerned using ABRicate (v 1.0.1) with CARD and PlasmidFinder. The plasmid contigs were extracted using SPAdes (v 3.14.4) in the plasmidSPAdes mode and then performed with alignment using the Basic Local Alignment Search Tool (BLAST, v 2.12.0+) server in NCBI.

### 2.5. Biofilm Formation Assessment

Biofilm formation assays were performed using the classical crystal violet method [28]. The experimental strains were cultured in the BHI medium for bacterial activation, and then 10^7^ CFUs of microbial suspensions were added to each well of the 96-well microtiter plate containing 200 μL of sterile seawater and incubated for 4 days at 25 °C. After incubation, the suspended bacteria were rinsed three times with PBS. The remaining cellular layers were stained with 1% crystal violet solution for 30 minutes and then washed with deionized water. The crystal violet bound to biofilm was dissolved in 95% ethanol, and the optical density was quantified at A595 using a microplate reader (BioTek, Washington, DC, USA). Multiple comparisons of nonparametric tests were performed using Statistical Product and Service Solutions (SPSS, v 26.0) (IBM, New Orchard Road Armonk, New York, NY, USA).

## 3. Results

### 3.1. Occurrence of Listeria Isolates and Molecular Subtyping of L. monocytogenes

Among the sand samples collected along the coast, a total of 26 *Listeria* isolates were obtained, representing an observed frequency of 3.38% (26/769). The 16 *L. monocytogenes* strains were isolated from 2.08% (16/769) of the samples in Bays B, D, and E, as well as Village V and Island I. Seven strains of *L. fleischmannii* were found in Bays B, C, and E, with a total prevalence of 0.91% (7/769) and prevailing in Bay C. Three strains of *L. aquatica* were sporadically found in Bay E, with a total frequency of 0.39% (3/769) (Figure 1B,C). 

The *L. monocytogenes* isolates belonged to two lineages (I and II) and four sequence types (ST87, ST121, ST35, and ST85). The prevalent ST87 strains were isolated in Bays B and D, Village V, and Island I. ST121 and ST35 strains were found in Bay E. ST85 isolates were found in Bay E and Village V.

### 3.2. The Biofilm Formation Ability of Listeria from Beach Sand

*Listeria* demonstrated to have biofilm capacity in seawater (Figure 1D). Notably, the *L. monocytogenes* ST87 isolates demonstrated significantly superior biofilm formation in seawater compared with other ST isolates of *L. monocytogenes*. Additionally, *L. fleischmannii* isolates, similar to the ST87 *L. monocytogenes* isolates, exhibited a similarly robust capacity for biofilm formation in seawater, surpassing ST121, ST35, and ST85 *L. monocytogenes*, as well as *L. aquatica*, in terms of biofilm formation ability.

All the *L. monocytogenes* isolates in this study were found to carry known biofilm-associated genes (*lmo0673*, *lmo2504*, *luxS*, and *recO*) as well as their regulation genes (*argA*, *Rli60*, *sigB,* and *mogR*). These genes exhibited genetic polymorphism in different sequence types of isolates. Within an identical sequence type, a consistent allelic profile was observed, and its polymorphism was found to be associated with the ST type (Figure 2).

### 3.3. Virulence Genes, Resistance Genes, and Plasmids of L. monocytogenes Isolates

A total of 182 virulence and resistance genes were analyzed across the isolates. All 16 *L. monocytogenes* isolates harbored a significant number of both virulence and resistance genes (Figure 2). Specifically, all the *L. monocytogenes* isolates from beach sand possessed LIP-1 (including *plc*, *actA*, *prfA*, *plcB*, *hly*, and *mpl*), *inlA* (full length), *inlB*, *inlC,* and *iap* but not LIPI-3. Notably, LIPI-4 was found in the ST87 isolates. The ST121 isolates contained stress survival islet 2 (SSI-2), whereas the ST35 and ST85 isolates contained stress survival islet 1 (SSI-1). Additionally, the ST121 isolates carried the cadmium resistance gene *cadA1C1*, whereas the ST35 isolates had the *cadA3C3* gene. No disinfectant resistance genes were detected in these isolates. According to the CARD database, all *L. monocytogenes* isolates inherently carried antibiotic-resistance genes *norB* (for fluoroquinolone resistance), *fosX* (for fosfomycin resistance), and *mprF* (for cationic peptide resistance).

The ST121 isolates of *L. monocytogenes* harbored a 57 kb plasmid. Ten plasmids sharing 100% identity and 100% coverage with this 57 kb plasmid were discovered in *L. monocytogenes*, *L. innocua*, and *L. welshimeri* in the NCBI database. These plasmids were distributed in the strains across various regions, including the United States, Italy, Australia, Poland, and Russia (Table 1). The plasmid contained 67 coding sequences (CDSs), encoding 10 transposase elements and numerous functional proteins, such as putative UV-damage repair protein UvrX, type II toxin–antitoxin system, CRISPR-associated protein Cas5, cadmium-transporting ATPase, copper-translocating P-type ATPase CopB, glycine/betaine ABC transporter GbuC, and NADH peroxidase Npx (Figure 3).

### 3.4. The Genetic Relationship of L. monocytogenes Isolates

In this study, the genome comparison was performed between 7 ST87 isolates from beach sand and 654 other isolates of *L. monocytogenes* (264 isolates from humans, 206 isolates from food and food-processing environments, 95 isolates from environments, and 89 isolates from other sources). The phylogenetic analysis showed a distinct trend toward regional clustering, with the sandy-derived isolates in this study exhibiting a closer evolutionary relationship with isolates from China than those from other countries. The isolates from clinical cases were distributed in the most evolutionary branch (Appendix A). The complete LIPI-4 was present in all ST87 isolates (*n* = 661) but absent in isolates from other STs (*n* = 1358) (Appendix A).

The genomes of 4 ST121 isolates of *L. monocytogenes* from beach sand in this study were compared with those of 1313 isolates from other regions (787 isolates from food and food-processing environments, 375 isolates from environments, 122 from humans, 8 isolates from animals and feed, and 21 isolates from unidentified sources). The sandy-derived ST121 isolates exhibited the closest genetic relationship to four isolates from three clinical cases that occurred between 2007 and 2008 in China, with 36–51 SNP differences (Appendix A). Furthermore, the complete SSI-2 was present in the majority of ST121 isolates (1307 out of 1317, 99.2%) but absent in the isolates from other STs (Appendix A).

When compared with the 29 ST35 isolates of *L. monocytogenes* from other sources, the 3 isolates from beach sand were closest to the 3 isolates from Dianchi Lake black-headed gulls in China, with 1–8 SNP differences (Appendix A). 

For the ST85 strains of *L. monocytogenes*, two isolates from beach sand exhibited a close phylogenetic relatedness to eight isolates (five isolates from food and food production environments, and three isolates from unidentified sources) from the USA, China, and Australia. These isolates displayed a minor genetic variation with each other, ranging from one to nine SNPs (Appendix A). The high conservation of this sequence type increases the difficulty of its traceability.

### 3.5. The General Genomic Features of L. fleischmannii and L. aquatica Isolates

*L. fleischmannii* exhibited a size range of 2.7–3.1 Mbp, slightly smaller than the size range of *L. monocytogenes* (2.8–3.2 Mbp). *L. fleischmannii* possessed a total of 6170 genes and 1659 core genes based on the 16 *L. fleischmannii* isolates, including 7 isolates from beach sand and 9 from other sources. The core genome and pan-genome of *L. fleischmannii* were 29.5% smaller and 40.8% larger than those of *L. monocytogenes* which had 2354 core genes and 4383 pan-genes according to a previous report [29]. Despite its smaller size, the *L. fleischmannii* genome possesses a greater number of accessory genes, contributing to species diversity and conferring competitive benefits to individual organisms.

*L. aquatica* displayed relatively small genome sizes, ranging from 2.6 to 2.7 Mbp. *L. aquatica* possessed a total of 3851 genes and 2063 core genes based on the five isolates of *L. aquatica,* including two isolates from beach sand and three from other sources. The core genome and pan-genome of *L. aquatica* were 12.4% and 12.1% smaller than those of *L. monocytogenes* according to a previous report [29]. In *L. aquatica*, over half of the genes in the pan-genome are classified as core genes, highlighting the relatively high conservation of the *L. aquatica* genome.

The number of virulence genes in *L. fleischmannii* and *L. aquatica* was obviously less than that in *L. monocytogenes* (Figure 2). *L. fleischmannii* harbored some virulence and resistance genes associated with adherence (*lap* and *fbpA*), intracellular survival (*oppA, clpc, clpp, fri,* and *pycA*), invasion (*recA*), peptidoglycan modification (*oatA*), regulation (*lhrC, lisK, lisR*, and *sigB*), cold resistance (*cspB* and *cspD*), low pH resistance (*lmo0796*), lysozyme resistance (*spoVGII*), nisin resistance (*liar*), and salt resistance (*gbuA*). *L. aquatica* contained the genes associated with adherence (*lap*), immune modulation (*tcsA)*, intracellular survival (*oppA, per, tig, iap*), regulation (*fur* and *sigB*), cold resistance (*lmo0866* and *yycG*), and nisin resistance (*liar*). *L. fleischmannii* harbored the fluoroquinolone resistance gene *norB*, while *L. aquatica* did not carry antibiotic-resistant genes. Furthermore, the plasmid was not found in *L. fleischmannii* and *L. aquatica* isolates in this study.

## 4. Discussion

Water-related illnesses impose a significant financial burden in California, with an annual cost of USD 3.3 million, attributed to coastal water pollution at just two beach locations [30]. Cholera outbreaks have been linked to aquatic environments in lakeside areas, as evidenced by a time-series analysis of 85565 cholera cases [31]. *Escherichia coli* O157:H7, a pathogenic bacterium responsible for hemorrhagic colitis and hemolytic uremic syndrome, has been detected in marine environments and is known to persist for up to a year in various water environments [32]. *L. monocytogenes* is an important foodborne pathogen that is capable of infecting both human beings and animals. In addition to food contamination, multiple transmission modes have been reported, such as contact with animals, exposure to contaminated water and environment, etc. [17,33,34]. Recreational water activities in coastal areas contaminated with fecal matter have been associated with gastrointestinal disease (>120 million cases) and respiratory disease (>50 million cases) worldwide each year [35,36]. A loggerhead sea turtle stranded on a beach on northern Italy’s coastline was infected with ST6 *L. monocytogenes*, leading to multiple internal lesions and a fatal infection [37]. Exposure to beach sand and recreational seawater contaminated with *L. monocytogenes* poses a potential threat to human and animal health. Therefore, investigating the contamination of *Listeria* in the marine environment, and elucidating the prevalence and molecular characteristics of *L. monocytogenes* in beach sand, are important for the evaluation of infection risk for humans, and would be helpful to prevent listeriosis in people contacting this environment.

In this pioneering investigation of *Listeria* from beach sand within a popular tourist destination, *Listeria* spp. were identified in 26 out of the 769 samples collected from the coastal environment, including four sequence-type isolates (ST87, 121, 35, and 85) of *L. monocytogenes* and two other species (*L. fleischmannii* and *L. aquatica*). These findings suggest that marine environments could be a potential reservoir for *Listeria* species. 

ST87 is a common sequence type of *L. monocytogenes* in raw seafood and fresh aquatic products in China [38,39]. It was also found to be the dominant sequence type associated with clinical infection in China and rarely reported in Europe or the United States [13,40,41]. In this study, the observed prevalence of ST87 isolates of *L. monocytogenes* in beach sand and their enhanced biofilm formation indicate a potential risk for the persistent contamination of aquatic products and food security. Notably, the regional heterogenicity of the ST87 isolates was observed, and the sandy isolates showed a closer association with other isolates previously found in China (150 isolates from humans and 84 isolates from food and environments). A majority of virulence genes, including LIPI-4, a cluster of six genes associated with transplacental and central nervous system infection, were found in 661 ST87 isolates from this study and NCBI database, including 276 isolates from the United States (156 isolates from food and marine environments, 46 isolates from humans, and 74 isolat+es from other sources). This finding is consistent with a previous study on the ST87 strain in China [40]. The colonization of ST87 *L. monocytogenes* in coastal ecosystems signifies potential health risks for people through recreational activities and contaminated seafood consumption.

ST121 strains of *L. monocytogenes* are predominant in food and have the demonstrated ability to persist within food-associated environments [26]. Interestingly, the four ST121 isolates obtained from beach sand were grouped together with four human isolates derived from infant cord blood (two isolates), placenta, and ascites [42]. Although ST121 isolate infections are not frequently reported in humans, the correlation between the ST121 isolates found in beach sand and the clinical strains needs further epidemiological investigation to be confirmed. Remarkably, these eight isolates all carried the *inlA* gene encoding a full-length InlA with identical amino acid sequence, while 97.1% (68/70) of ST121 strains possessed a truncated *inlA* gene [43].

The plasmids within *L. monocytogenes* could augment the survival and virulence of LM populations [44,45]. A conserved 62 kb plasmid (pLM6179) was previously identified in 81.4% (57/70) of ST121 isolates and was found to contribute to resistance to salinity, oxidative stress, and acid environments [43]. In this study, a 57 kb plasmid was identified in ST121 strains, which was also found in diverse host strains in many countries between 1994 and 2020. This plasmid encodes multifunctional, and as yet uncharacterized proteins, such as UvrX for resistance to UV-induced damage, GbuC for resistance to osmosis, Npx for resistance to oxidative stress, the type II TA system for plasmid maintenance, and transposases that facilitate genomic islands across a range of bacteria [45,46,47]. Therefore, it might contribute to the tolerance of *L. monocytogenes* against sunlight exposure and sea salt in beach environments.

Seabirds can carry pathogenic microbes and serve as a long-distance disseminator of infectious agents such as *L. monocytogenes* through migration [19]. According to annual waterbird surveys, a total of 2899 black-headed gulls was recorded in this region between 2008 and 2020 [48]. Our study found that the ST35 isolate (LM_E1) in beach sand had the closest association with the two isolates from black-headed gulls (Accession Number: GCA_009788365.1 and GCA_009788295.1) in 2016 with one SNP difference in *lmo0651* (coding transcriptional regulator), suggesting the possibility that these isolates originated from the same source and therefore highlighting the potential for long-distance spread meditated by black-headed gull migration. Fecal contamination with *L. monocytogenes* dropped in the sand might result in the persistence of the pathogen in recreational marine environments, thus leading to an infection risk for people around beaches.

The bacterial community in the seawater and river environments is subjected to geographical location, with greater species abundance observed downstream, potentially attributable to diverse factors such as human activity and algal blooms [49,50]. In this study, five distinct bays along the coastline of a city in China were investigated and denoted as A, B, C, D, and E. Each of these bays has distinct traits and varying levels of *Listeria* detection. No *Listeria* spp. was detected in samples of Bay A, which was located upstream and had the cleanest coastal water and the least population density among these bays. Bay B, approximately 10 kilometers from the downtown area, featured pristine white sand beaches and had the isolation of *L. monocytogenes* and *L. fleischmannii*. Bay C, a niche tourism hotspot enriched with coral reefs, had the highest isolation of *L. fleischmannii* and was located approximately 4 kilometers away from the downtown area. Bay D, located in the downtown area with a public bathing beach of visa-free access, had a relatively high isolation rate of *L. monocytogenes*. Bay E, located downstream of the other bays, had a relatively high species diversity of *Listeria*. Additionally, Island I and Village V, both popular leisure resorts located between Bay A and Bay B, were contaminated with *L. monocytogenes* and offered various maritime entertainments and sports for tourists. Our finding suggests that the prevalence of *L. monocytogenes* increased with the degree of human activity, in agreement with studies conducted in Danish aquatic and fish-processing environments [51].

Biofilms of marine bacteria are ubiquitous in marine ecosystems, facilitating rapid adaptation to environmental challenges and participating in a set of functions [52]. In this study, all *Listeria* isolates could generate biofilm in seawater, with the *L. monocytogenes* ST87 and *L. fleishmanii* isolates showing particularly robust biofilm formation. *L. monocytogenes* isolates harbored many biofilm formation genes with genetic polymorphism, while *L. fleischmannii* isolates may have yet unidentified biofilm formation genes, thus warranting further study. Additionally, the presence of genes resistant to environmental adaptation, especially those related to salt resistance (*gbuABC* and *opuCABC*) and UV resistance (*uvrX*), likely enhances the survival and proliferation of *L. monocytogenes* within marine environments. The continuous emergence of seafood contamination by *L. monocytogenes* has been well documented, with 0.5–1.0 % of listeriosis cases associated with the consumption of ready-to-eat seafood [53]. Furthermore, water-based recreational activities also pose an exposure risk for people regarding *L. monocytogenes* infection through accidental ingestion.

Environmental surveys of *L. monocytogenes* on the central California coast showed a prevalence of 41.9% to 62% in public access surface water, with serotype 4b strains accounting for most of the isolates [20,21,22]. Further analysis revealed that 90% of the *L. monocytogenes* isolates from the central California coast contain the *inlA* gene encoding for intact InlA, which differs significantly from the isolates found in foods and food-processing environments [54]. Another study on *L. monocytogenes* isolates in Morocco showed a prevalence of 2.5% in the marine environment, with serotype 1/2b strains in the majority [23]. The presence of *L. monocytogenes* in a variety of marine habitats presents a potential contamination source and a threat to human health.

As an accumulation substrate and source of bacterial pollution from diverse origins, beach sand could serve as a significant reservoir for bacterial pathogens, with 10–100-fold greater bacterial abundance than that of adjacent recreational waters [55]. Studies reported *L. monocytogenes* contamination in foods in 1998 and 2014–2015 in this coastal region [56,57]. There is a possibility of cross-contamination between marine environments and food-related environments. Our findings demonstrate that *Listeria* can exist and produce biofilm in marine environments, and the potential persistent colonization would cause seafood contamination and, consequently, human infection. The continuous monitoring of pathogenic bacteria, including *L. monocytogenes*, in these environments is crucial for public health in coastal cities. Susceptible individuals should avoid engaging in beach recreational activities or exposing their wounds to marine environments.

## 5. Conclusions

Our study first reveals the genetic and phenotypic profiles of diverse species isolates of *Listeria* from the coastal environment in China, highlighting the potential risk of *L. monocytogenes* infection in the marine environment. Notably, ST87 *L. monocytogenes* isolates emerged as the predominant sequence-type strains in recreational sand beaches, exhibiting robust capacity for biofilm formation in seawater. This finding highlights marine environments as significant reservoirs for the virulent strains of *L. monocytogenes*. However, this study was a preliminary exploration and had limited sample coverage of the entire coastal environment; thus, the sampling range and sample size should be expanded for a more comprehensive risk assessment of *L. monocytogenes* in a future study. The routine surveillance of pathogens in highly populated coastal regions and marine fishing areas is important for public health. Effective epidemic monitoring by public health authorities is essential to evaluate the potential risk of *L. monocytogenes* infection in marine environments and to develop strategies for listeriosis prevention for visitors and residents in coastal regions.

## Figures and Tables

**Figure 1 microorganisms-11-02166-f001:**
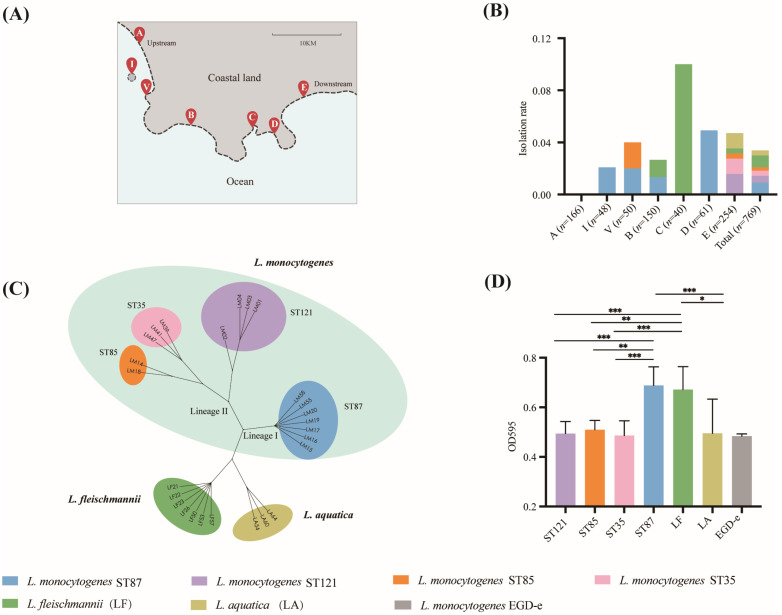
(**A**) The sample sites on the seabeach of the coastal city; (**B**) isolation rate of *L. monocytogenes*, *L. fleischmannii*, and *L. aquatica* in each site; (**C**) unrooted phylogenetic tree based on core genes of *Listeria* isolates collected in this study (*n* = 26); (**D**) biofilm formation of *Listeria* in TSB and seawater (* *p* < 0.05, ** *p* < 0.01, and *** *p* < 0.001).

**Figure 2 microorganisms-11-02166-f002:**
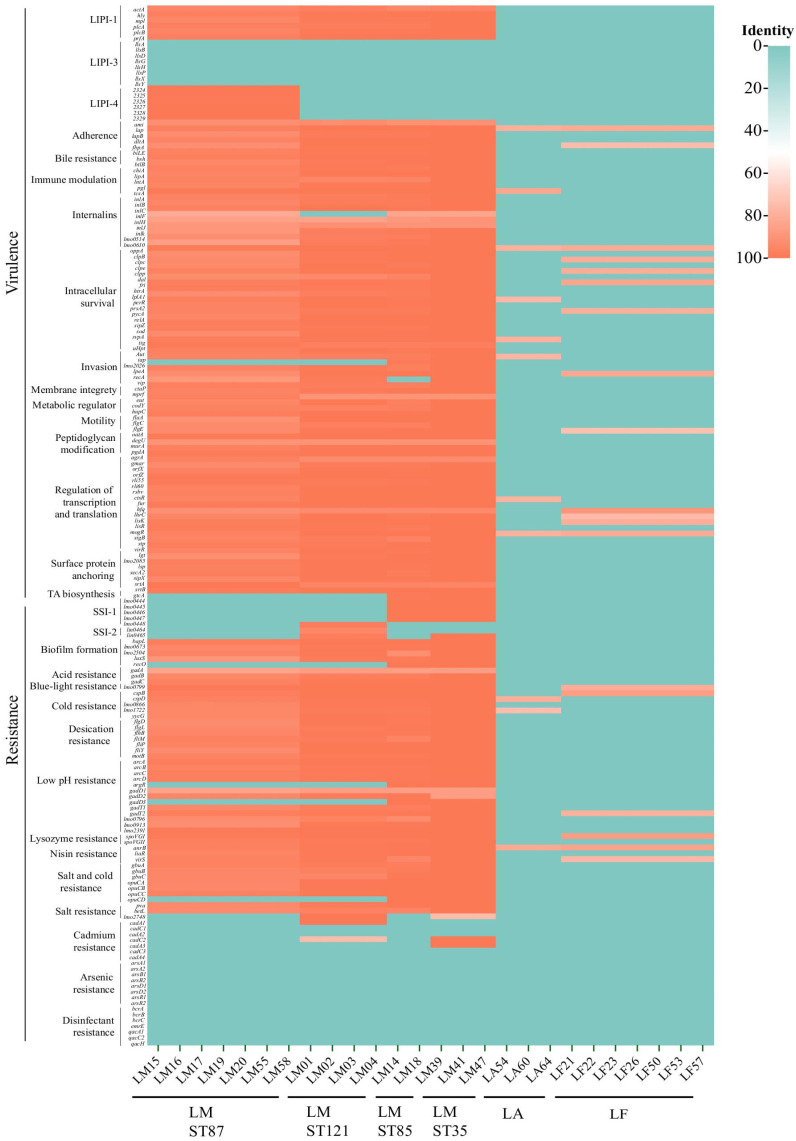
Color heatmap of presence (orange)/absence (green) gene matrix with the 182 virulence and resistance genes constructed by the *Listeria* strains isolated in this study.

**Figure 3 microorganisms-11-02166-f003:**
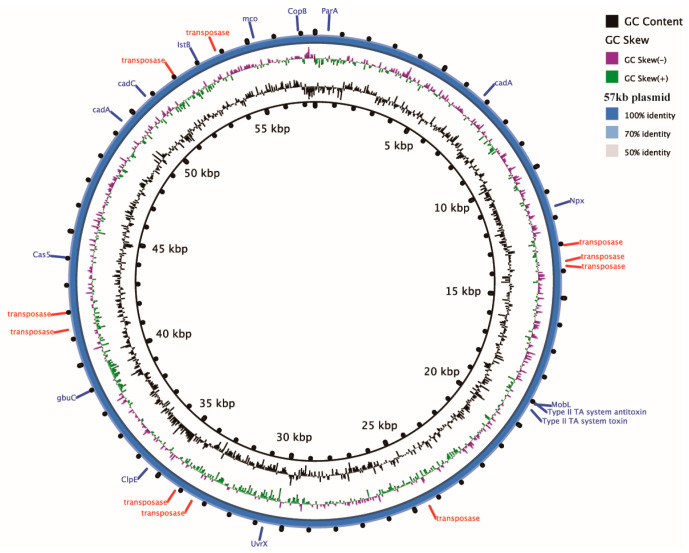
The ring image of 57 kb plasmid. BRIG was used to generate a visual representation of 57,530 nt contig. GC contents and skews are presented in the inner rings. The annotation of this plasmid with the known genes indicated is represented in the outer ring.

**Table 1 microorganisms-11-02166-t001:** The information of plasmids from *Listeria* isolates in other countries.

Accession No. of Plasmid	Species	ST	Accession No. of Strains	Geographic Location	Isolation Source	Collection Date	Assemble Level
FR667692.1	* LM	66	GCA_000197755.2	unknown	unknown	unknown	**
CP006595.1	* LM	3	GCA_000438585.1	USA	food	1994	**
CP015985.1	* LM	7	GCA_001596775.2	Italy	blood (human)	2015	**
CP025561.1	* LM	199	GCA_003031975.1	unknown	unknown	unknown	**
CP041214.1	* LM	3	GCA_004142705.2	USA	chocolate milk	2018	**
CP045973.1	* LM	122	GCA_009664775.1	Australia	human	2009	**
CP090056.1	* LM	31	GCA_021403065.1	USA	Salmon processing facility	1998	**
MZ127848.1	* LIN			Poland	food contact surface swab	2009	**
CP045744.1	* LIN		GCA_009648575.1	Italy	minced meat	2005	**
MZ869809.1	* LW			Russia	environmental surface, meat processing plant	2020	**

* LM: Listeria monocytogenes; LIN: Listeria innocua; LW: Listeria welshimeri; ** complete genome.

## Data Availability

The genome data is presented in Appendix A.

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
