# Peer review of "The Isolation, Genetic Analysis and Biofilm Characteristics of Listeria spp. from the Marine Environment in China"

_microorganisms, 2023, doi:10.3390/microorganisms11092166_

Round 1
Reviewer 1 Report
Introduction -
1replace even fatal infections — with “other”
2. Some grammar changes needed.
methods
1. several abbreviations have been used for the first time without full forms.please correct
discussion
1. one major— can the authors report the incidence and prevalence of listeria at the location studied?
2 any recent outbreaks in the area?
Moderate quality, needs editing prior to publication
Author Response
Response to Reviewer 1 Comments
Manuscript ID: microorganisms-2527988
Type of manuscript: Article
Title of Article: Isolation, genetic analysis and biofilm characteristics of Listeria spp. from marine environment in China
Dear Reviewer,
Thank you very much for your valuable comments and constructive suggestions. Based on the comments, we have revised the relevant parts in the manuscript, and check that all references are relevant to the contents of the manuscript. Additionally, some errors and deficiencies are also revised through our self-check process. For each specific comment, we have made point-by-point reply as follows. The “italic” text denotes the reviewers’ comments, the “red” text indicates our responses. A revised manuscript with the correction sections was red marked for easy check. We hope that the revised manuscript is acceptable for further review.
We really appreciate your help.
Sincerely yours,
Changyun Ye
Comments and Suggestions of Reviewer:
Point 1: Introduction part: replace even fatal infections — with “other”
Response 1: Thank you for your suggestions. We have revised the sentence in the article as follows "L. monocytogenes can cause septicemia, meningitis, abortion, and other complications in human and animals "(Line 31-33).
Point 2: Introduction part: Some grammar changes needed.
Response 2: Thank you for the comment. We have made the necessary grammar changes to the entire introduction part (Line 28, 38, 40, 43-45, 47-49, 51-54, 56-59).
Point 3: Method part: several abbreviations have been used for the first time without full forms. please correct.
Response 3: Thank you for the suggestion. We have supplemented the full forms of the abbreviations when introduced for the first time (Line 76, 86-87, 97-98, 102-103, 118, 127-128).
Point 4: Discussion part: one major— can the authors report the incidence and prevalence of listeria at the location studied?
Response 4: Thank you for your suggestion. We have supplemented this information in the main text (Line 351-354). Currently, there is no comprehensive monitoring of the incidence and prevalence of Listeria in this coastal province. However, there have been two studies conducted in the past that provide relevant information on the occurrence of Listeria in the region. The first study conducted in 1998 found a high contamination rate (40% - 47.6%) in Listeria in animal-based foods (Wen Ruirong, 1999, Chin J Health Lab Technol) (The literature before 2000 may not have a DOI number in China). The second study conducted in 2014-2015 found that L. monocytogenes was detected in frozen meat and frozen fish with prevalence rates of 0.9% and 1.2% (DOI:10.3969/j.issn.1006-3110.2018.03.002. in Chinese). Both of studies did not conduct molecular epidemiology and genomic characterization of Listeria, thereby precluding direct molecular comparison analysis with our research. While there have been limited reports in certain regions, additional research will be conducted to gain a better understanding of the Listeria situation at this region in the future study.
Point 5: Discussion part: any recent outbreaks in the area?
Response 5: Thank you for your suggestion. The comprehensive surveillance system for clinical L. monocytogenes has not been established in China. According to a systematic review of human listeriosis of China in 1964-2010, there were no outbreak in this coastal province (DOI: 10.1111/tmi.12173 IF3.3 Q1). The lack of recent outbreak reports may be attributed to several factors. One significant factor is the insufficient awareness and understanding of Listeria infections among local healthcare professionals in this underdeveloped area. Additionally, the clinical cases may not have been effectively monitored and recorded, leading to a dearth of available data on recent outbreaks.
Despite the absence of reported outbreaks in our study area, we acknowledge the importance of continuous vigilance and research on Listeria infections. As part of our commitment to contributing to public health, we will continue to conduct further investigations to better comprehend the prevalence and potential risks of Listeria in the region.
-----End of Reply to Reviewer #1------
Reviewer 2 Report
The manuscript by Pan et al. is well written and conducted, having an original contribution to the knowledge of the epidemiology, genetic structure and biofilm characteristics of Listeria spp. isolated from marine environment. I support its further processing after appropriate modifications as outlined below:
L3: „spp.” and throughout the manuscript– not italic
L13 vs. L15 – please clarify and revise throughout the manuscript: (ST) 87 or (ST)87?
L29: please insert the appropriate reference for the LPSN explaining the meaning of this acronym
L29: „Listeria monocytogenes is the main human pathogenic....” – please carefully document and mention the public health potential of other Listeria spp.
L45: please insert an appropriate reference after the end of the sentence
L49: „[8, 9]” – please avoid the using of space delimitation between the references (according to MDPI guideline)
L52: at the end of the sentence: the documentation of the importance of L. monocytogenes for the food industry can be improved by consulting and citing recently published valuable articles (e.g. https://doi.org/10.4315/JFP-21-172)
L58-66: the paragraph can be condensed in a single complex sentence, and the detailed information can be used within the discussion section
L73: the reviewer wonder, in the lack of any previously conducted investigation in China and concerning the total number of the processed samples (n=769), was any sample size calculation based on statistic tools taken into consideration?
L80: “(Oxoid)” – according to the MDPI journal requirements for all the mentioned reagents and equipment please uniformly mention the production company name, city, and country origin
L83: “gene prs” – please explain/justify the reason of the choosing of this type of gene for molecular confirmation
L86: the amount of the used Listeria culture for DNA extraction in unclear. Please specify!
L123: “SPSS” – please mention the meaning of this acronym
L127: “3.38%” – when you express overall prevalence values, please indicate in brackets the value of the 95% Confidence Interval
L137: “ST85),” – please replace the comma with point
L164: “We identified” – please avoid the using of personal mode verbs formulations throughout the manuscript, it is not so characteristic for the scientific style
Overall, the results and discussions are well presented!
L340: “conclusions” – sentence case
Within the conclusion section the authors must highlight the importance of study results for public health authorities, the study limitations and further strategies is the approached research area
The reference list is not in agreement with the journal requirement! Please carefully revise it!
Author Response
Response to Reviewer 2 Comments
Manuscript ID: microorganisms-2527988
Type of manuscript: Article
Title of Article: Isolation, genetic analysis and biofilm characteristics of Listeria spp. from marine environment in China
Dear Reviewer,
Thank you very much for your valuable comments and constructive suggestions. Based on the comments, we have revised the relevant parts in the manuscript, and check that all references are relevant to the contents of the manuscript. Additionally, some errors and deficiencies are also revised through our self-check process. For each specific comment, we have made point-by-point reply as follows. The “italic” text denotes the reviewers’ comments, the “red” text indicates our responses. A revised manuscript with the correction sections was red marked for easy check. We hope that the revised manuscript is acceptable for further review.
We really appreciate your help.
Sincerely yours,
Changyun Ye
Comments and Suggestions of Reviewer:
The manuscript by Pan et al. is well written and conducted, having an original contribution to the knowledge of the epidemiology, genetic structure and biofilm characteristics of Listeria spp. isolated from marine environment. I support its further processing after appropriate modifications as outlined below:
Thank you for your positive review of our manuscript. We have tried our best to revise and enhance the manuscript's quality according to your comments.
Point 1: L3: „spp.” and throughout the manuscript– not italic
Response 1: We feel sorry for the incorrect use of italics with ‘spp.’. We have made the necessary adjustments throughout the manuscript (Line 3, 10-11, 70, 86, 99, 101, 259, 314).
Point 2: L13 vs. L15 – please clarify and revise throughout the manuscript: (ST) 87 or (ST)87?
Response 2: Thank you for your reminder. We feel sorry about the inconsistency in the description. "ST87" is the accurate description based on referenced literature (DOI: 10.1128/JCM.41.2.757-762.2003 IF9.4 Q1; DOI: 10.1186/s12864-019-6399-1 IF4.4 Q1; DOI: 10.1016/j.ijfoodmicro.2019.03.016 IF5.4 Q1). We have revised the content to ensure consistency (Line 13).
Point 3: L29: please insert the appropriate reference for the LPSN explaining the meaning of this acronym
Response 3: Thank you for the suggestion, we have inserted the appropriate reference (DOI: 10.1093/nar/gkt1111 IF14.9 Q1) (Line 29).
Point 4: L29: „Listeria monocytogenes is the main human pathogenic....” – please carefully document and mention the public health potential of other Listeria spp.
Response 4: Thank you for your suggestion. We have documented and emphasized the public health potential of these other Listeria spp. in our revised manuscript: “L. monocytogenes can cause septicemia, meningitis, abortion, and other complications in human and animals, while L. ivanovii causes similar pathological manifestations in ruminants, except for meningoencephalitis. Despite limited availability of clinical and pathogenicity data for other species, investigating the ecology, genetics, and survival strategies of these species is crucial for future public health preparedness” (Line 31-36).
Point 5: L45: please insert an appropriate reference after the end of the sentence
Response 5: Thank you for your suggestion, we have inserted the appropriate reference (DOI: 10.1038/ng.3501 IF30.8 Q1; DOI: 10.1038/nmicrobiol.2016.185 IF28.3 Q1) (Line 48, 50).
Point 6: L49: „[8, 9]” – please avoid the using of space delimitation between the references (according to MDPI guideline)
Response 6: Thank you for your reminder. we have revised the reference format throughout the manuscript according to MDPI guideline using Endnote.
Point 7: L52: at the end of the sentence: the documentation of the importance of L. monocytogenes for the food industry can be improved by consulting and citing recently published valuable articles (e.g. https://doi.org/10.4315/JFP-21-172IF: 2.0 Q3)
Response 7: Thank you for your comment, we have inserted the appropriate reference (DOI: 10.4315/JFP-21-172 IF: 2.0 Q3) (Line 56).
Point 8: L58-66: the paragraph can be condensed in a single complex sentence, and the detailed information can be used within the discussion section
Response 8: Thank you for your valuable suggestion. We have condensed the paragraph into a single complex sentence, allowing for a clearer and more concise description (line 61-64). The detailed information was elaborated upon in the discussion section as recommended (line340-348).
Point 9: L73: the reviewer wonder, in the lack of any previously conducted investigation in China and concerning the total number of the processed samples (n=769), was any sample size calculation based on statistic tools taken into consideration?
Response 9: Thank you for your valuable suggestion. In this study, we did not perform a statistical sample size calculation due to the lack of prior investigations in China. Instead, we selected seven sampling locations and initially set a minimum of 100 samples per bay. We expanded or reduced the sample size based on the coastline length, area, and accessibility of each collection location. According to our findings, the observed frequency was 3.38% (26/769), suggesting that approximately 3,000 samples would be required for a more precise estimation. However, due to the limitation such as sampling points interval, sample availability, sample transportation, and limited resources (including manpower and funding), we had no chance to further expand the sample size in this study. Therefore, the 769 samples were collected as comprehensively, considering the coastline length and openness of each bay to achieve a balanced representation across different regions. In the future studies, we will employ statistical sampling methods to provide a robust and representative dataset, which would help us to get more detailed and exact result about the distribution of L. monocytogenes in the region.
Point 10: L80: “(Oxoid)” – according to the MDPI journal requirements for all the mentioned reagents and equipment please uniformly mention the production company name, city, and country origin
Response 10: Thank you for the comment. We have revised the method section with the complete information (Line 75, 76-77, 79, 80-81, 86-87, 89, 91-92, 126-127).
Point 11: L83: “gene prs” – please explain/justify the reason of the choosing of this type of gene for molecular confirmation
Response 11: Thank you for your suggestion. Gene prs is specific for Listeria genus and we have inserted the appropriate reference (DOI: 10.1128/JCM.42.8.3819-3822.2004 IF9.4 Q1) (Line 84).
Point 12: L86: the amount of the used Listeria culture for DNA extraction in unclear. Please specify!
Response 12: Thank you for the suggestion. We have clarified that the DNA extraction were performed from the entire bacterial culture obtained from one BHI agar plate after incubating for 18 hours (Line 86-89).
Point 13: L123: “SPSS” – please mention the meaning of this acronym
Response 13: Thank you for the suggestion. We have supplemented the meaning of the acronym "SPSS" in the manuscript (Line 127-128).
Point 14: L127: “3.38%” – when you express overall prevalence values, please indicate in brackets the value of the 95% Confidence Interval
Response 14: Thank you for the comment. We feel sorry for the lack of clarity in the initial presentation of the results. The calculation of 3.38% is based on 26 out of 769 isolates of L. monocytogenes obtained from the sand samples collected along the coast. We have made the necessary change, replacing "prevalence" with "observed frequency," to provide a more accurate description of our results. The calculation of a 95% Confidence Interval is not applicable in this context. The manuscript has been revised to clarify this point, as indicated in the main text (Line 131-132).
Point 15: L137: “ST85),” – please replace the comma with point
Response 15: Thank you for the suggestion. We apologize for the oversight in the use of a comma instead of a point. We have made the correction (Line142).
Point 16: L164: “We identified” – please avoid the using of personal mode verbs formulations throughout the manuscript, it is not so characteristic for the scientific style
Response 16: Thank you for the comment. We sincerely appreciate the reviewer's guidance on adhering to a more scientific style throughout the manuscript. We have revised full text of the manuscript to avoid using personal mode verbs formulations (Line 131-132, 169-171, 183, 193-194, 259, 266-267, 311-312, 314).
Point 17: Overall, the results and discussions are well presented!
Response 17: Thank you for the positive review of our work. We appreciated for your encouragement and help.
Point 18: L340: “conclusions” – sentence case
Response 18: Thank you for the suggestion. We have revised it to “Conclusions” (Line 360).
Point 19: Within the conclusion section the authors must highlight the importance of study results for public health authorities, the study limitations and further strategies is the approached research area
Response 19: Thank you for your valuable suggestions. We appreciate your guidance and have revised the conclusion section by providing a clearer overview of the study's implications and limitations (Line 366-374).
We summarize the limitations of this study, and it was a preliminary exploration with limited sample coverage of the entire coastal environment in China. The sampling range and sample size should be expanded for more comprehensive risk assessment of L. monocytogenes in the future study. Additionally, effective epidemic monitoring by public health authorities is essential to evaluate the potential risk of L. monocytogenes infection in marine settings and to develop preventive strategies, ensuring the health and safety of visitors and residents in coastal regions.
Point 20: The reference list is not in agreement with the journal requirement! Please carefully revise it!
Response 20: Thank you for your comment. We feel sorry for the incorrect reference list and have revised the reference format throughout the manuscript.
-----End of Reply to Reviewer #2------
Round 2
Reviewer 1 Report
Thank you for making suggested changes
Minor corrections needed
Author Response
Response letter
Manuscript ID: microorganisms-2527988
Type of manuscript: Article
Title of Article: Isolation, genetic analysis and biofilm characteristics of Listeria spp. from marine environment in China
Dear Reviewer,
Thank you very much for your valuable comment. We have revised and enhanced the English language quality of the manuscript. Additionally, sample site map and detailed description of each location have been presented. All quality control and quality assurance procedures were checked and added more information. For each comment, we have made point-by-point reply as follows. The “italic” text denotes the comments, the “red” text indicates our responses. A revised manuscript with the correction sections was blue marked for easy check. We hope that the revised manuscript is acceptable for further review.
We really appreciate your help.
Sincerely yours,
Changyun Ye
Reviewer #1
Point 1: Minor corrections needed
Response 1: Thank you for your suggestion. We have tried our best to make some corrections of the text and enhance the English language quality of the paper (Line 6, 10, 14, 18, 65, 72-73, 82-83, 85-89, 91, 98-100, 112, 125-126, 141-145, 152, 164, 182-183, 192-193, 202, 228-230, 254-255, 258, 267, 269, 274, 277, 287, 291, 311, 325, 341-349, 368, 387). Additionally, we have added a visual representation of the sample sites on the sea beach to show the geographical relationships between seven locations (Figure 1A, Line 142-146).
